# HybridoNet-Adapt: A domain-adapted framework for accurate lithium-ion battery RUL prediction

**Khoa Tran**[1], **Bao Huynh**[1], **Tri Le**[1], **Lam Pham**[2], **Vy-Rin Nguyen**[3], **Duong Tran Anh**[4,5], **Hung-Cuong Trinh**[6]*

1 AIWARE Limited Company, Da Nang, Vietnam, 2 AIT Austrian Institute of Technology GmbH, Vienna, Austria, 3 Software Engineering Department, FPT University, Da Nang, Vietnam, 4 Laboratory of Environmental Sciences and Climate Change, Institute for Computational Science and Artificial Intelligence, Van Lang University, Ho Chi Minh City, Vietnam, 5 Faculty of Environment, Van Lang School of Technology, Van Lang University, Ho Chi Minh City, Vietnam, 6 Natural Language Processing and Knowledge Discovery Research Group, Faculty of Information Technology, Ton Duc Thang University, Ho Chi Minh City, Vietnam

* trinhhungcuong@tdtu.edu.vn

**Data availability statement:** - TRI dataset is available from the public database of Toyota Research Institute (https://data.matr.io/1/projects/5c48dd2bc625d700019f3204). - LHP

## Abstract

Accurate prediction of the Remaining Useful Life (RUL) of lithium-ion batteries is critical for safe, reliable Battery Health Management in diverse operating conditions. Existing RUL models often fail to generalize when test data diverge from the training distribution. To address this, we introduce HybridoNet-Adapt, a domain-adaptive RUL prediction framework that explicitly bridges the gap between labeled source and unlabeled target domains. During training, we minimize the Maximum Mean Discrepancy (MMD) between feature distributions to learn domain-invariant representations. Simultaneously, we employ two parallel predictors—one tailored to the source domain and one to the target domain—and balance their outputs via two learnable trade-off parameters, enabling the model to dynamically weight domain-specific insights. Our architecture couples this adaptation strategy with LSTM, multi-head attention, and Neural ODE blocks for deep temporal feature extraction, but its core novelty lies in the MMD-based alignment and hybrid prediction mechanism. On two large, publicly available battery datasets, HybridoNet-Adapt consistently outperforms non-adaptive baselines (Structural Pruning, Multi-Time Scale Feature Extraction Hybrid model, XGBoost, Elastic Net), archiving an RMSE reduction of up to 152 cycles under domain shifts. These results demonstrate that incorporating domain adaptation into RUL modeling substantially enhances robustness and real-world applicability.

## 1 Introduction

### 1.1 Motivations

Lithium-ion batteries (LIBs) [1], renowned for their affordability and high energy density, are extensively utilized [2–5] in electric vehicles (EVs), portable devices, and energy storage stations. The global lithium-ion battery (LIB) market is projected to surpass 170 billion dollars

dataset is available from the Mendeley Data repository (DOI: 10.17632/nsc7hnsg4s.2, https://data.mendeley.com/datasets/nsc7hnsg4s/2).

**Funding:** Hung-Cuong Trinh, Ton Duc Thang University, No Grant number. The funder does not play any role in the study design, data collection and analysis, decision to publish, or preparation of the manuscript.

**Competing interests:** The authors have declared that no competing interests exist.

by 2030 [6]. With the wide-ranging adoption of LIBs, interest in battery health management (BHM) has surged within both academia and industry in recent years. In a BHM system, several common and essential techniques are employed, including thermal management [7,8], fault diagnosis/detection [9], state of charge (SOC) [10–12] and state of health (SOH) [13] estimation, remaining useful life (RUL) prediction [14,15], and cycle life early prediction [16–19]. Among these, RUL prediction plays a crucial role in ensuring the proactive maintenance, minimizing downtime, and enhancing the operational efficiency of LIBs over their lifespan. RUL can be assessed based on the number of remaining cycles the battery can undergo before reaching its EOL. RUL prediction falls into three categories: model-based, data-driven, and hybrid approaches.

Model-based approaches often utilize physics-based degradation models, such as the Double Exponential Model (DEM) [20], two-phase degradation models [21], and Markov Models [22], constructed using early-cycle data (200-500 cycles) to forecast the entire battery's capacity degradation curve. However, relying solely on maximum discharge capacity degradation during early cycles often leads to inaccuracies due to the influence of various factors (current, voltage, temperature, time) and sudden changes in degradation trends [20,21].

Data-driven models predict the RUL of LIBs by analyzing data during current cycles. Techniques like dual-input Deep Neural Networks (DNN) [23], 1D Convolutional Neural Networks (1DCNN) [24], Dense layers [25], Long Short-Term Memory (LSTM) networks [26], and Echo State Networks (ESN) [27] have shown superior performance.

Hybrid approaches combine model-based and data-driven methods to improve RUL prediction. For instance, a hybrid model using the Double Exponential Degradation Model (DEDM) and Gated Recurrent Unit (GRU) network fused with a Bayesian neural network (BNN) offers enhanced predictions [28].

Accurately predicting the RUL of LIBs remains a significant challenge due to the complex, non-linear, and cycle-dependent degradation behavior of battery features. This variability necessitates a highly adaptive prediction model capable of tracking and learning degradation patterns over time throughout charge–discharge cycles. The specific challenges associated with this task will be discussed in the following section.

## 1.2 Problem statement and potential

As summarized in Table 1, current state-of-the-art studies categorize RUL prediction methods into two primary approaches: historical data-independent methods, which estimate the current RUL based on current and preceding few cycles, and historical data-dependent methods, which leverage early-cycle data to predict the battery's full lifespan.

Historical data-dependent methods estimate the future capacity trajectory. The EOL cycle index is then determined, typically defined as 70% [33] or 80% [20] of the nominal capacity, and the RUL is calculated as the difference between the current cycle index and the EOL cycle index. These approaches primarily rely on early-cycle data. While historical data-dependent methods can achieve reasonable accuracy in benchmark evaluations [28,38], they face practical limitations such as the unavailability of early-cycle records, varying operational conditions [17] throughout a battery's lifespan, and challenges in battery repurposing [43]. Therefore, historical data-independent approaches are considered more suitable for real-world scenarios.

Small datasets, like the Oxford Battery dataset (13 cells) and NASA battery datasets (4–34 cells) limit model robustness in real-world failure prediction. In contrast, large datasets such as the TRI dataset (124 cells, fast-charging) and the LHP dataset (77 cells, diverse discharge)

**Table 1**. Overview of RUL prediction methods in LIB research.

| Method Type | Prediction Target | Reference | Dataset | Signal Preprocessing | Prediction Model |
|---|---|---|---|---|---|
| Historical data-independent | RUL | [23] | LHP 2022 [29] | Feature-based condition extraction; sequential feature sampling | Dual-input DNN |
| | RUL | [15] | Oxford Battery [30] | Ageing-correlated parameter extraction | Physics-based DNN |
| | RUL | [31] | TRI [16], another[32] | Feature extraction, normalization | Encoding Net |
| Historical data-dependent | Capacity | [33] | NASA 2016 [34] | CEEMDAN | Single-input PA-LSTM |
| | Capacity | [21] | NASA 2007 [35] | Binary segmentation + particle filtering | Two-phase capacity degradation model |
| | Capacity | [36] | CALCE-CX2, CALCE-CS2 [37] | EMD, GRU-FC | CNN predicting max discharge capacity |
| | Capacity | [38] | NASA PCoE, CALCE [37] | Variational Mode Decomposition (VMD), KL divergence | Bayesian-optimized LSTM, ESN |
| | Capacity | [28] | NASA PCoE, Random Walk [39] | Z-score normalization, EMD | GRU-CNN, DEDM, BNN |
| | Capacity | [40] | NASA PCoE [41], TRI [16] | Feature extraction of temperature, current, voltage; sliding window denoising | Cascaded forward neural network (CFNN) |
| | Capacity | [42] | TRI [16] | Feature extraction | MLP Encoder, TransConv Encoder |
| | RUL | [20] | TRI [16] | Spline interpolation (length 1000) | CNN, DEM, GPR |

provide extensive charge-discharge scenarios, making them well-suited for both training and testing of data-independent models.

Signal preprocessing can be limited by high dimensionality, especially with variational decomposition methods like EMD and VMD, which preserve or expand the original signal size. In contrast, statistical feature extraction methods—such as mean and standard deviation—offer a low-dimensional, efficient alternative that captures essential characteristics, making them ideal for real-time industrial applications.

Model-based and hybrid approaches typically rely on early-cycle data for RUL prediction, yet each battery exhibits unique degradation patterns over its lifespan requiring adaptive data-driven strategies. In data-driven approaches, domain adaptation (DA) techniques such as domain-adversarial neural networks (DANN)[44,45] and generative adversarial networks (GANs)[46] offer effective solutions for transferring degradation patterns from a source domain to improve prediction in the target domain. [31] proposed a transferable RUL prediction method using DA that enforces cycle-consistency of degradation trends across batteries, aligning feature representations to mitigate domain shifts. Their approach improves cross-battery generalization, it relies on comparable degradation levels between source and target domains. This work demonstrates that domain adaptation techniques benefit RUL prediction in the battery domain, highlighting an direction for further exploration. To address these mentioned challenges and leverage the identified potential, our proposed approach is introduced in the next section.

## 1.3 Main contribution

The main contributions of this work are summarized as follows:

- We propose a historical data-independent RUL prediction framework for lithium-ion batteries that relies solely on current and recent cycling data, eliminating the need for early-cycle information. The prediction model integrates advanced deep learning components—including Long Short-Term Memory (LSTM), Multihead Attention, and Neural Ordinary Differential Equations (NODE) blocks—as a powerful feature extractor, along with linear

layers in the predictors. Furthermore, a domain adaptation strategy is employed, combining two predictors with trainable trade-off parameters and an MMD-based loss to learn domain-invariant features, thereby enhancing transferability from source to target domains. This strategy improves the generalizability of the prediction model.

- The framework includes a lightweight yet robust preprocessing pipeline-noise reduction, statistical feature extraction (e.g., mean, standard deviation), and normalization—to improve signal quality and reduce dimensionality for efficient, real-time prediction.

- Extensive evaluations on the two largest publicly available datasets of A123 APR18650M1A cells [16,23], covering diverse charging and discharging conditions, validate the superior performance of our approach in real-world battery health management.

The remainder of this paper is organized as follows. Sect 2 introduces the preliminaries. Sect 3 presents the proposed method. Sect 4 describes the experiments and discussion. Finally, Sect 5 concludes the paper and outlines future work.

## 2 Preliminaries

This section presents an overview of the key components of the prediction model architecture, including LSTM, Multihead Attention, and NODE blocks.

### 2.1 LSTM

The LSTM network [26] is a recurrent architecture designed to mitigate the vanishing gradient problem by introducing gating mechanisms. Its operations at time step $t$ are defined as:

$$i_t = \sigma(W_i x_t + U_i h_{t-1} + b_i), \tag{1}$$

$$f_t = \sigma(W_f x_t + U_f h_{t-1} + b_f), \tag{2}$$

$$o_t = \sigma(W_o x_t + U_o h_{t-1} + b_o), \tag{3}$$

$$\tilde{c}_t = \tanh(W_c x_t + U_c h_{t-1} + b_c), \tag{4}$$

$$c_t = f_t \odot c_{t-1} + i_t \odot \tilde{c}_t, \tag{5}$$

$$h_t = o_t \odot \tanh(c_t), \tag{6}$$

where $x_t \in \mathbb{R}^{d_x}$ is the input vector at time step $t$. $h_t \in \mathbb{R}^{d_h}$ and $c_t \in \mathbb{R}^{d_h}$ are the hidden and cell states, respectively. $i_t, f_t, o_t \in \mathbb{R}^{d_h}$ denote the input, forget, and output gates. $\tilde{c}_t \in \mathbb{R}^{d_h}$ is the candidate cell state. $W_* \in \mathbb{R}^{d_h \times d_x}$, $U_* \in \mathbb{R}^{d_h \times d_h}$, and $b_* \in \mathbb{R}^{d_h}$ are trainable weight matrices and bias vectors. $\sigma(\cdot)$ is the sigmoid activation, $\tanh(\cdot)$ is the hyperbolic tangent, and $\odot$ denotes element-wise multiplication.

### 2.2 Multihead attention

Multihead Attention [47] is a critical mechanism in Transformer models, enabling the network to attend jointly to information from different subspaces. The basic building block is the scaled dot-product attention:

$$\text{Att}(Q, K, V) = \text{softmax}\left(\frac{QK^\top}{\sqrt{d_k}}\right)V, \tag{7}$$

where $Q \in \mathbb{R}^{n \times d_k}$, $K \in \mathbb{R}^{n \times d_k}$, and $V \in \mathbb{R}^{n \times d_v}$ denote the query, key, and value matrices, respectively, and $d_k$ is the dimensionality of the keys. In a multihead setting, multiple attention heads are computed as

$$h_i = \text{Att}(QW_i^Q, KW_i^K, VW_i^V), \quad i = 1, \dots, h, \tag{8}$$

where $W_i^Q \in \mathbb{R}^{d_{\text{model}} \times d_k}$, $W_i^K \in \mathbb{R}^{d_{\text{model}} \times d_k}$, and $W_i^V \in \mathbb{R}^{d_{\text{model}} \times d_v}$ are trainable projection matrices. Finally, the outputs of all heads are concatenated and linearly transformed:

$$\text{MHA}(Q, K, V) = \text{Concat}(h_1, \dots, h_h)W^O, \tag{9}$$

where $W^O \in \mathbb{R}^{hd_v \times d_{\text{model}}}$ is a trainable output projection matrix, and $h$ is the number of attention heads.

## 2.3 NODE

NODE is a framework that extends deep learning architecture by modeling continuous-time dynamics instead of discrete transformations between layers. In NODE, the evolution of a hidden state $h(t)$ is governed by an ordinary differential equation (ODE):

$$\frac{dh(t)}{dt} = f(h(t), t, \theta), \tag{10}$$

where $f$ is a neural network parameterized by $\theta$. The final state $h(t) \in \mathbb{R}^{d_h}$ is the hidden state at time $t$, obtained by solving this ODE over a time interval, which provides a flexible and memory-efficient representation.

# 3 Proposed method

## 3.1 Overall architecture

Fig 1 illustrates the RUL prediction process for Lithium-ion battery cells. In the data collection phase, Lithium Iron Phosphate (LFP)/graphite cells are monitored to capture voltage, current, and capacity signals for each individual charge-discharge cycle.

Regarding cycle life degradation, the cycle life of a battery is defined as the total number of charge-discharge cycles from the Beginning of Life (BOL) to the End of Life (EOL). The EOL is typically identified when the battery's maximum capacity in a charge-discharge cycle degrades to 70% [33] or 80% [20] of its nominal capacity. The RUL, expressed in terms of the remaining number of cycles, is computed as:

$$\text{RUL} = N_{\text{life}} - N_{\text{age}}, \tag{11}$$

where $N_{\text{life}}$ is the total cycle life of the battery, $N_{\text{age}}$ is the number of aging cycles already completed.

In the signal preprocessing phase, the raw signals—voltage, current, and capacity from the most recent charge-discharge cycles—are passed through a noise-reduction filter named median filter [48] to smooth out sudden peaks. The filtered signals are then processed using feature extraction methods, including mean, standard deviation (Std), minimum (Min), maximum (Max), variance (Var), and median (Med) [49,50]. The extracted features for each cycle

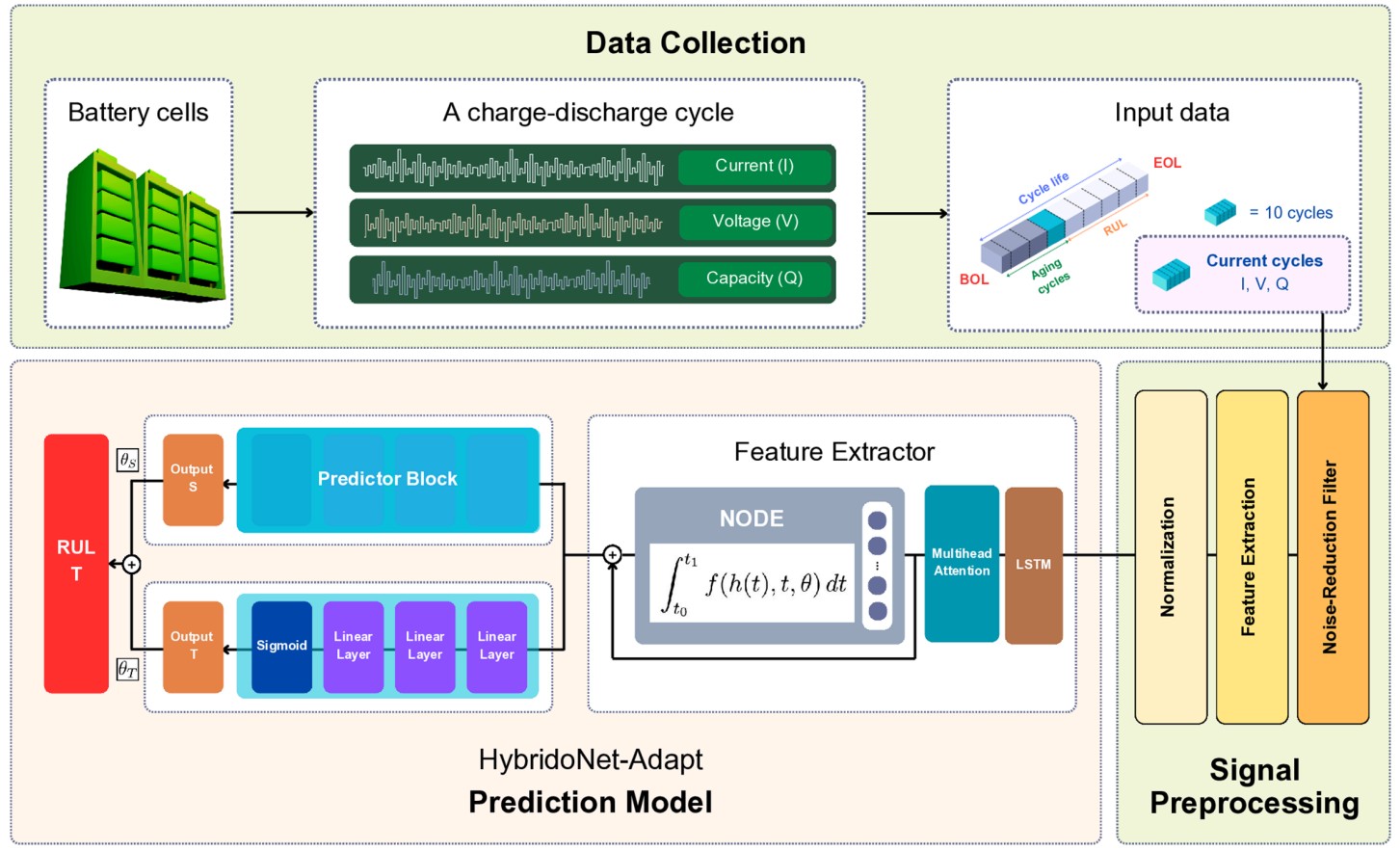

**Fig 1. The overall RUL prediction process for Lithium-ion battery cells.**

are represented as

$$X^i = \left[ x^i_{\text{current}}, x^i_{\text{voltage}}, x^i_{\text{capacity}} \right], \quad X^i \in \mathbb{R}^{3 \times 6}, \tag{12}$$

where the 3 correspond to the three signal types (voltage, current, and capacity), and the 6 represent the extracted features (Mean, Std, Min, Max, Var, Med). Each input sample to the prediction model consists of 10 selected cycles, uniformly sampled from a 30-cycle window (i.e., one cycle every three cycles) [29]. The input sample is represented as

$$\mathbf{X^i} = \left[ X^1, X^2, ..., X^{10} \right], \quad \mathbf{X} \in \mathbb{R}^{10 \times 3 \times 6}. \tag{13}$$

Thus, the shape of the total target input data after the feature extraction step becomes

$$\mathbf{X^T} \in \mathbb{R}^{N \times 10 \times 3 \times 6}, \tag{14}$$

where $N$ denotes the number of samples. During normalization, a MinMaxScaler is fitted and applied to scale each feature across all time steps and samples between 0 and 1.

In the prediction phase, the prediction model, named **HybridoNet-Adapt**, maps the target input $\mathbf{X^T}$ to the predicted RUL $\mathbf{Y^T} \in \mathbb{R}^{N \times 1}$. The details of the proposed RUL prediction model are presented in the following section.

## 3.2 HybridoNet-Adapt: A proposed RUL prediction model with novel domain adaptation

As shown in Fig 2, HybridoNet-Adapt is composed of four key components: the source predictor $G_Y^S$, the target predictor $G_Y^T$, and the feature extractor $G_F$, which is equipped with a DA technique to bridge the distribution gap between the source and target domains.

The feature extractor integrates a LSTM (Sect 2.1), a Multihead Attention mechanism (Sect 2.2), and a NODE block (Sect 2.3). The NODE block models the hidden state $h(t)$, which evolves continuously over time according to the following ODE: $\frac{dh(t)}{dt} = f(h(t), t, \theta)$, where $h(t)$ denotes the hidden state at time $t$, $f$ is a trainable function parameterized by $\theta$, and $t$ represents the continuous time variable. In our implementation, $f$ is a single linear layer to strike a balance between performance and computational efficiency. The initial condition for the NODE block is given by $h(t_0)$, and the final transformed state $h(t_1)$ is obtained by solving the ODE over the time interval $[t_0, t_1]$:

$$h(t_1) = h(t_0) + \int_{t_0}^{t_1} f(h(t), t, \theta)\, dt. \tag{15}$$

In our experiments, the time bounds are set to $t_0 = 0$ and $t_1 = 1$, based on empirical results (see Fig 9). The function $h(t)$ thus represents the dynamic trajectory of the hidden state under continuous transformation, enabling the model to capture nuanced temporal dependencies.

The target and source predictions in HybridoNet-Adapt are respectively computed as follows:

$$\hat{Y}_i^T(X_i) = \theta^S G_Y^S\big(G_F(X_i)\big) + \theta^T G_Y^T\big(G_F(X_i)\big), \tag{16}$$

$$\hat{Y}_i^S(X_i) = G_Y^S\big(G_F(X_i)\big), \tag{17}$$

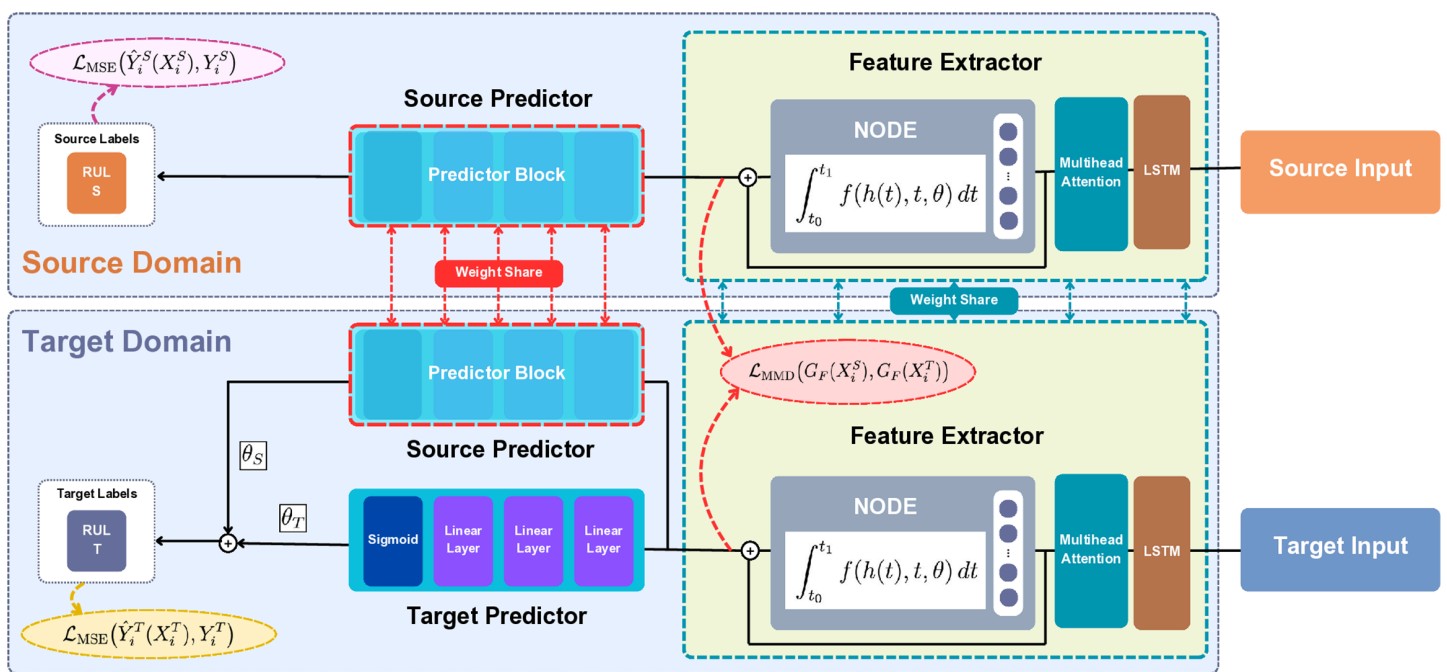

**Fig 2. Architecture of the proposed HybridoNet-Adapt model during training process with domain adaptation.**

where $\theta^S$ and $\theta^T$ are learnable trade-off parameters that balance the contributions from the source and target predictors. The outputs $\hat{Y}_i^T$ and $\hat{Y}_i^S$ denote the target and source predictions, respectively. The source predictor $G_Y^S$ and the feature extractor $G_F$ are trained using both source and target data, enabling the model to transfer domain-invariant features from the source domain to the target domain. This promotes robust prediction performance in the target domain, especially in scenarios where the model would otherwise underperform if trained solely on target data.

The hyperparameters of the proposed HybridoNet-Adapt are summarized in Table 2. LayerNorm denotes layer normalization [51], and Dropout refers to a dropout layer [52] with a rate of 0.1. FC represents a fully connected layer [53], while in the NODE block, $h(t)$ is parameterized by an FC layer. The learnable trade-off parameters $\theta^S$ and $\theta^T$ are initialized at 0.5 and updated during training. ReLU refers to the rectified linear unit activation function [54], Sigmoid refers to the sigmoid activation function [55], and BN denotes batch normalization [56]. The columns Input Dim. and Output Dim. specify the dimensionality of the inputs and outputs of each module, respectively. These values are determined empirically, as discussed in Sect 4.7. Note: The output from the Multihead Attention block is taken from the last step along the time dimension.

To optimize the model, we employ a domain adaptation strategy that combines two loss functions: the mean squared error (MSE) loss, $\mathcal{L}_{\text{MSE}}$, used for regression targets $\hat{Y}_i^T$ and $\hat{Y}_i^S$; and the maximum mean discrepancy (MMD) [57] loss, $\mathcal{L}_{\text{MMD}}$, which encourages alignment between the feature distributions extracted from the source and target domains. The total loss function is defined as:

$$\mathcal{L}(X_i^S, X_i^T, Y_i^S, Y_i^T) = \mathcal{L}_{\text{MSE}}\big(\hat{Y}_i^S(X_i^S), Y_i^S\big)$$
$$+ \mathcal{L}_{\text{MSE}}\big(\hat{Y}_i^T(X_i^T), Y_i^T\big)$$
$$+ \lambda\, \mathcal{L}_{\text{MMD}}\big(G_F(X_i^S), G_F(X_i^T)\big), \tag{18}$$

where $X_i^S$ and $X_i^T$ are the input samples from the source and target domains, respectively, and $Y_i^S$ and $Y_i^T$ are their corresponding RUL labels. The hyperparameter $\lambda$ controls the weight of the MMD loss in the overall objective. The MMD loss quantifies the discrepancy between the distributions of source and target feature embeddings. Given extracted feature sets $F_i^S$ from the source domain $G_F(X_i^S)$ and $F_j^T$ from the target domain $G_F(X_j^T)$, it is defined as:

$$\mathcal{L}_{\text{MMD}}(F_i^S, F_j^T) = \frac{1}{n^2}\sum_{i=1}^{n}\sum_{j=1}^{n} k(F_i^S, F_j^S) + \frac{1}{m^2}\sum_{i=1}^{m}\sum_{j=1}^{m} k(F_i^T, F_j^T)$$
$$- \frac{2}{nm}\sum_{i=1}^{n}\sum_{j=1}^{m} k(F_i^S, F_j^T), \tag{19}$$

**Table 2. Hyperparameters of HybridoNet-Adapt.**

| Module | Layer/ Block | Input Dim. | Output Dim. |
|---|---|---|---|
| $G_F(\cdot)$ | 2× LSTM (hidden = 128, layers = 2, dropout = 0.1) + LayerNorm | $10 \times 18$ | $10 \times 128$ |
| | Multihead Attention (4 heads) + LayerNorm | $10 \times 128$ | 128 |
| | NODE Layer (FC) + LayerNorm | 128 | 128 |
| $G_Y^S(\cdot)$ | FC + ReLU + BN + Dropout(0.1) | 128 | 64 |
| | FC + ReLU + BN + Dropout(0.1) | 64 | 32 |
| | FC + Sigmoid | 32 | 1 |
| $G_Y^T(\cdot)$ | FC + ReLU + BN + Dropout(0.1) | 128 | 64 |
| | FC + ReLU + BN + Dropout(0.1) | 64 | 32 |
| | FC + Sigmoid | 32 | 1 |

where $k(\cdot, \cdot)$ is a kernel function, commonly chosen as the Gaussian kernel: $k(x, y) = \exp\left(-\frac{\|x-y\|^2}{2\sigma^2}\right)$, with $\sigma$ as the kernel bandwidth parameter. $n$ and $m$ denote the number of training samples from source and target domains, respectively. The MSE loss is used to optimize the regression outputs by penalizing the squared differences between predicted and ground truth values. It is defined as:

$$\mathcal{L}_{\text{MSE}}(\hat{Y}_i, Y_i) = \frac{1}{u} \sum_{i=1}^{u} \left(\hat{Y}_i - Y_i\right)^2, \tag{20}$$

where $\hat{Y}_i$ denotes the predicted value, and $Y_i$ is the corresponding label for the RUL. $u$ is the number of training samples.

In the following section, a series of experiments are conducted to identify the optimal configuration of HybridoNet-Adapt and to demonstrate its superiority over state-of-the-art methods.

For validating the proposed domain adaptation used in HybridoNet-Adapt, we construct a supervised learning model named *HybridoNet*, consisting of the target predictor $G_Y^T$ and the feature extractor $G_F$. This model is trained solely on labeled target data using the MSE loss function. By comparing between HybridoNet (without domain adaptation) and HybridoNet-Adapt (with domain adaptation), we highlight the performance improvements achieved through the incorporation of our proposed Domain Adaptation technique.

## 4 Experiments and discussion

### 4.1 Experimental setup

Our proposed RUL model is implemented using the PyTorch framework and optimized using the AdamW algorithm [58] to minimize the respective loss functions. All experiments are conducted on an NVIDIA A100 GPU with 80GB of memory. Each experiment is trained for 10 epochs with a batch size of 128 and a fixed learning rate of 0.0005. To reduce variability in the training process, each experiment is repeated 10 times, and the final prediction is computed as the average of these runs. The training data is divided into 90% for training and 10% for validation, with the model selected based on the lowest RMSE on the validation set (see Sect 4.3). The weighting factor $\lambda$ in Eq 18 is dynamically adjusted during training using the following schedule [59]:

$$\lambda = \frac{2}{1 + e^{-10 \cdot \frac{\text{epoch}}{\text{epochs}}}} - 1, \tag{21}$$

where epoch denotes the current training epoch, and epochs is the total number of training epochs.

### 4.2 Datasets

**4.2.1 First dataset: Varied fast-charging conditions, with consistent discharging conditions.** The first dataset, referred to as the TRI dataset [16], encompasses a detailed study of 124 LFP/graphite lithium-ion batteries. Each LIB in the dataset has a nominal capacity of 1.1 Ah and a nominal voltage of 3.3 V. The cycle life span of these batteries ranges from 150 to 2,300 cycles, showcasing a wide spectrum of longevity. In terms of operational conditions, all LIBs were subjected to uniform discharge protocols. Specifically, they were discharged at a constant current rate of 4 C until the voltage dropped to 2 V, followed by a constant voltage discharge at 2 V until the current diminished to C/50. The LIBs were charged at rates between

3.6 C and 6 C, under a controlled temperature of 30°C within an environmental chamber. The dataset contains approximately 96,700 cycles, making it one of the largest datasets to consider various fast-charging protocols. The dataset is divided into three distinct parts: a training set with 41 LIBs, a primary test set with 43 LIBs, and a secondary test set comprising 40 LIBs.

**4.2.2 Second dataset: Varied discharge conditions, with consistent fast-charging conditions.** The second dataset, referred as the LHP dataset [29], was developed through a battery degradation experiment involving 77 cells (LFP/graphite A123 APR18650M1A) with a nominal capacity of 1.1 Ah and a nominal voltage of 3.3 V. Each of the 77 cells was subjected to a unique multi-stage discharge protocol, while maintaining an identical fast-charging protocol for all cells. The experiment was conducted in two thermostatic chambers at a controlled temperature of 30°C. The dataset encompasses a total of 146,122 discharge cycles, making it one of the largest datasets to consider various discharge protocols. The cells exhibit a cycle life ranging from 1,100 to 2,700 cycles, with an average of 1,898 cycles and a standard deviation of 387 cycles. The discharge capacity as a function of cycle number reveals a wide distribution of cycle lives. The dataset is divided into two distinct parts: a training set with 55 LIBs, and a test set with 22 LIBs.

## 4.3 Evaluation metrics

To evaluate RUL prediction, we use Root Mean Square Error (*RMSE*) [60], R-squared ($R^2$) [38,61], and Mean Absolute Percentage Error (*MAPE*) [62]. These are calculated as follows:

$$RMSE(y_i, \hat{y}_i) = \sqrt{\frac{1}{n} \sum_{i=1}^{n} (y_i - \hat{y}_i)^2}, \tag{22}$$

$$MAPE(y_i, \hat{y}_i) = \frac{1}{n} \sum_{i=1}^{n} \frac{|y_i - \hat{y}_i|}{y} \times 100, \tag{23}$$

$$R^2(y, \hat{y}) = 1 - \frac{\sum_{i=1}^{n} (y_i - \hat{y}_i)^2}{\sum_{i=1}^{n} (y_i - \bar{y})^2}. \tag{24}$$

Where $y_i$ and $\hat{y}_i$ are the observed and predicted RUL, respectively. $y$ is cycle life. The smaller the *RMSE* and *MAPE*, and the larger the $R^2$, the better the performance.

## 4.4 Signal analysis

Fig 3a and 3b analyze battery cycle life. Fig 3a tracks an individual cell's charge and discharge capacities, marking EOL when the maximum capacity degrades to 80% of nominal capacity. Fig 3b compares cycle life across cells, revealing significant variation in discharge capacity. This variability challenges prediction models for RUL, emphasizing the need for accurate and adaptable RUL predictions for BHM systems.

## 4.5 Signal preprocessing

Before feature extraction step in the signal preprocessing phase (as mentioned in Sect 3), the raw signals exhibit sudden peaks and fluctuations, resembling noise. Smoothing the time-series data can help reduce noise and enhance the key characteristics of the signal. To achieve

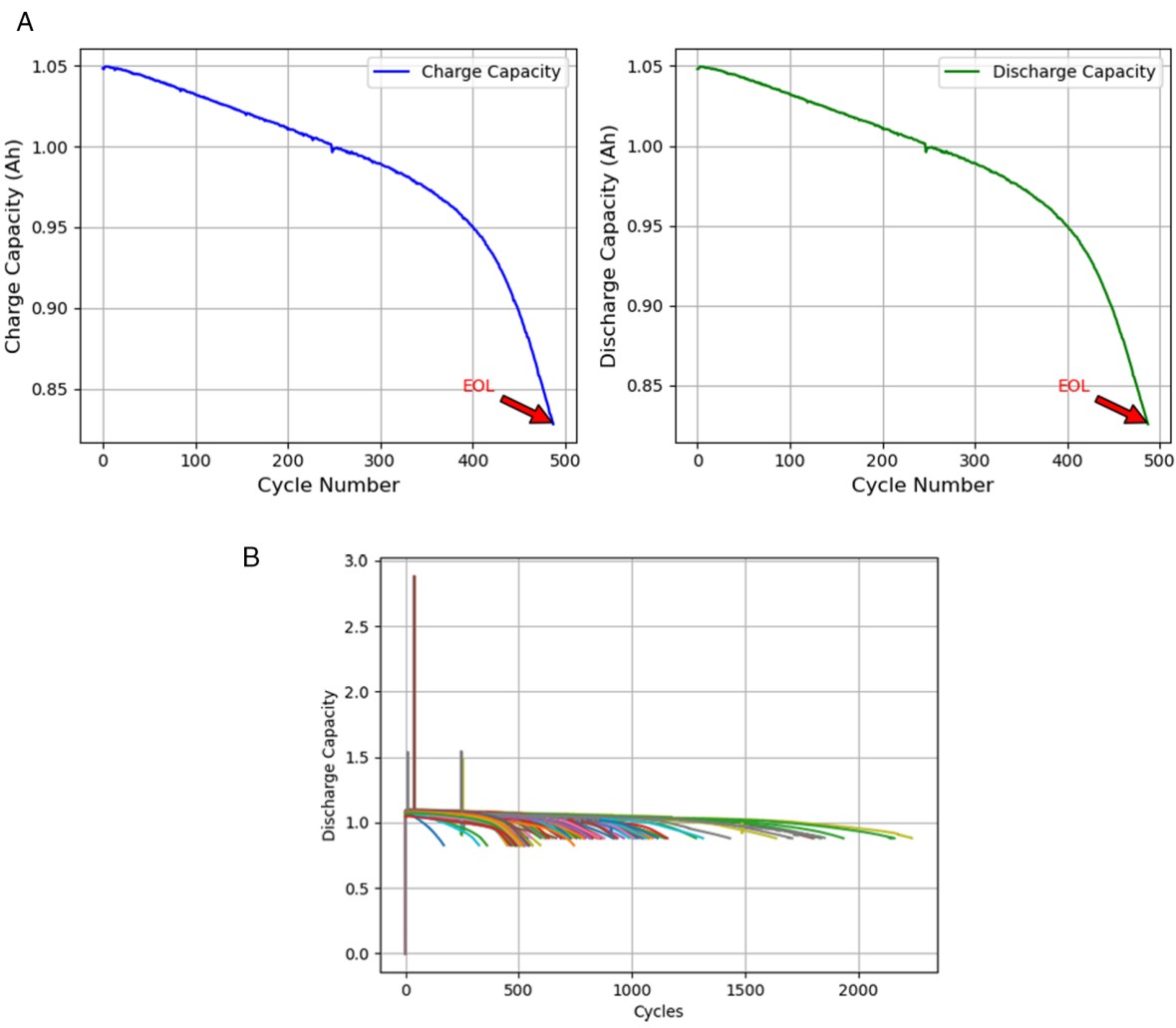

**Fig 3. Comparison of maximum charge and discharge capacities over cycle life for battery cells.** (a) Maximum charge and discharge capacities over charge-discharge cycles for a single battery cell. (b) Maximum discharge capacities over charge-discharge cycles for many battery cells.

this, a median filtering method is applied to eliminate abrupt peaks in the signals before feature extraction. As a result, the application of median filtering improves overall model performance. The filtered data leads to better RMSE, $R^2$, and MAPE (%) values compared to the unfiltered data, as illustrated in Fig 4.

Fig 5a presents the feature importance ranking derived from XGBoost [63] trained and evaluated on the second dataset. The analysis considers 19 common feature extraction methods: 75th, 90th, 50th, 25th, and 10th percentiles; Maximum; Range; Energy; Mean; Interquartile Range (IQR); Median; Skewness; Standard Deviation (Std); Kurtosis; Root Mean Square (RMS); Minimum; Variance; Zero-Crossing Rate; and Autocorrelation (implemented using NumPy [64] and SciPy [65]). Among these, the 75th, 90th, and 10th percentiles demonstrate the highest contribution. We group the feature extraction methods into five categories: three based on high importance (Groups 1–3), one consisting of fundamental statistics (Group 4), and one hybrid group (Group 5):

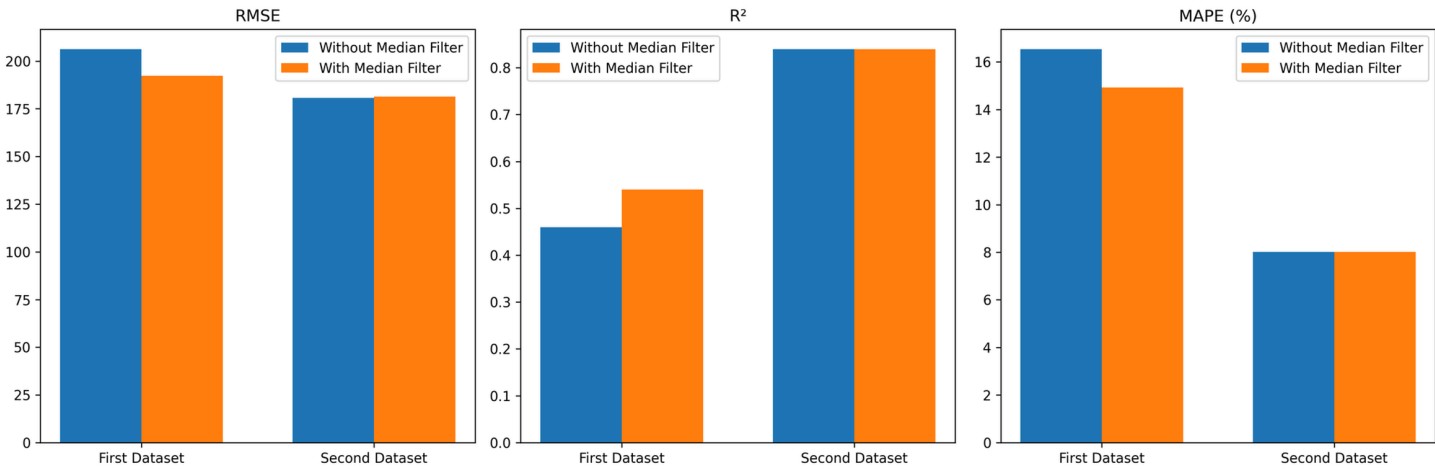

**Fig 4. Comparison of RUL prediction performance with and without median filtering.**

- **Group 1:** Percentiles 75th, 90th, 10th, Range, IQR.
- **Group 2:** Group 1 + Maximum, Percentiles 25th, Skewness, Kurtosis, Standard Deviation (Std).
- **Group 3:** Group 2 + Energy, Median, RMS, Mean, Variance, Minimum.
- **Group 4:** Mean, Std, Min, Max, Variance, Median.
- **Group 5:** Group 4 + Percentiles 75th, 90th, 10th.

Fig 5b shows that although Group 1 and 2 contain features with the highest importance scores, they do not yield strong overall prediction performance. Group 4 (fundamental statistics), which is selected as the feature extraction step during the Signal Preprocessing phase of the proposed framework, combines both high- and low-importance features but achieves the best performance, with an RMSE of 181.45, significantly outperforming others.

### 4.6 Feature extractor

The feature extractor is progressively developed, starting with an LSTM architecture and sequentially integrating Multihead Attention (MA) and a NODE block. To evaluate the effectiveness of each component, we assess the performance of HybridoNet-Adapt at different stages. With each addition as shown in Fig 6, the model's predictive capability improves. Ultimately, HybridoNet-Adapt achieves an RMSE of 166.33, an $R^2$ score of 0.86, and a MAPE of 7.44%, demonstrating its superior performance.

### 4.7 HybridoNet-Adapt with domain adaption

HybridoNet-Adapt is evaluated with various feature loss functions, including CORAL Loss, Domain Loss [44], MMD, as well as combinations such as MMD with Domain Loss, and MMD with Domain Loss and CORAL Loss, as shown in Fig 7. The results indicate that using only MMD as the feature loss function yields the best performance, achieving an RMSE of 160.05.

To determine the optimal hyperparameters, including hidden dimension of all layers, the number of recurrent LSTM layers, and the dropout rate, 27 experiments were conducted. The results are presented in Fig 8. In the graph, *L* represents the number of recurrent layers, *H*

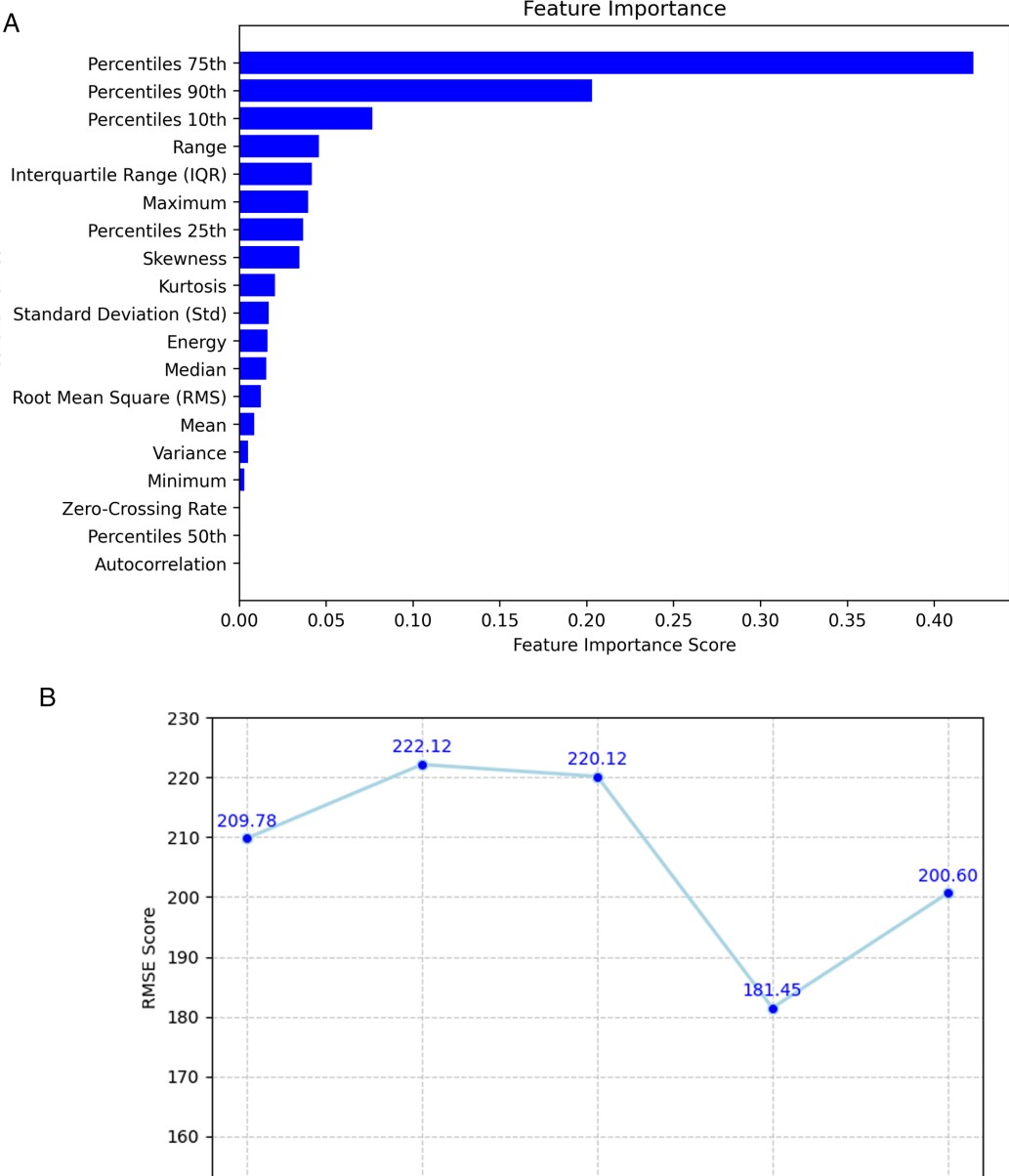

**Fig 5. Feature contribution analysis: (a) feature importance ranking, and (b) RMSE comparison across feature extraction groups.**

denotes the hidden dimension size. Based on RMSE score, the best performance is achieved with 2 recurrent LSTM layers, a hidden dimension of 64, and a dropout rate of 0.1.

To identify the optimal time step in the sequence dimension for both Multihead Attention and NODE outputs, a comprehensive evaluation was conducted. Various NODE output time steps ranging from 2 to 6 were tested, along with different Multihead Attention output time step selections, including the last, the second-to-last, and the mean time step. As shown

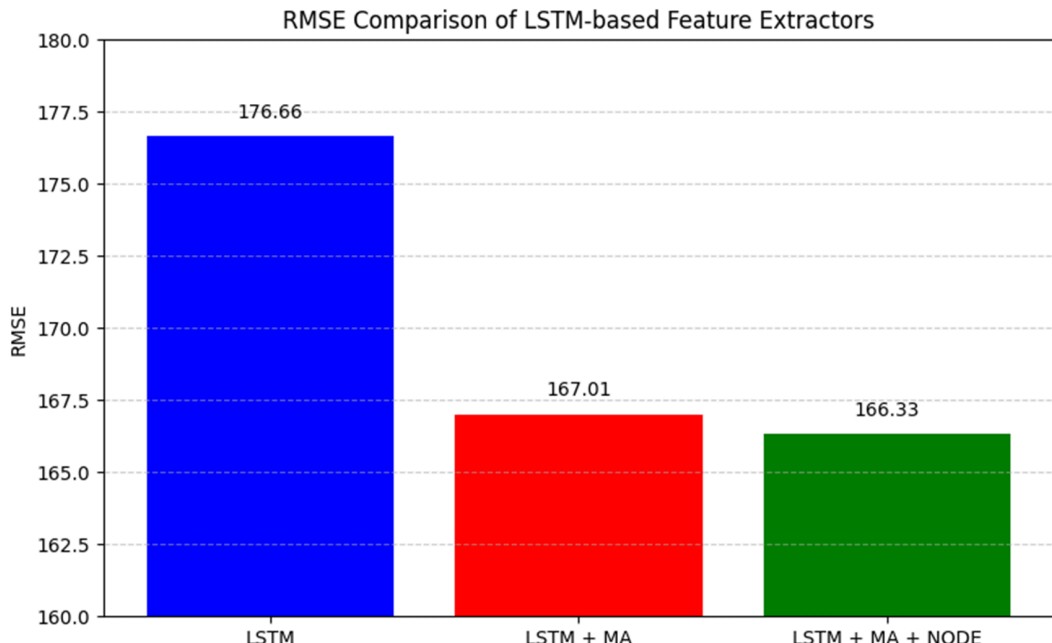

**Fig 6. Performance comparison of LSTM-based blocks for feature extractor.** Experiment on the testing data of the second dataset.

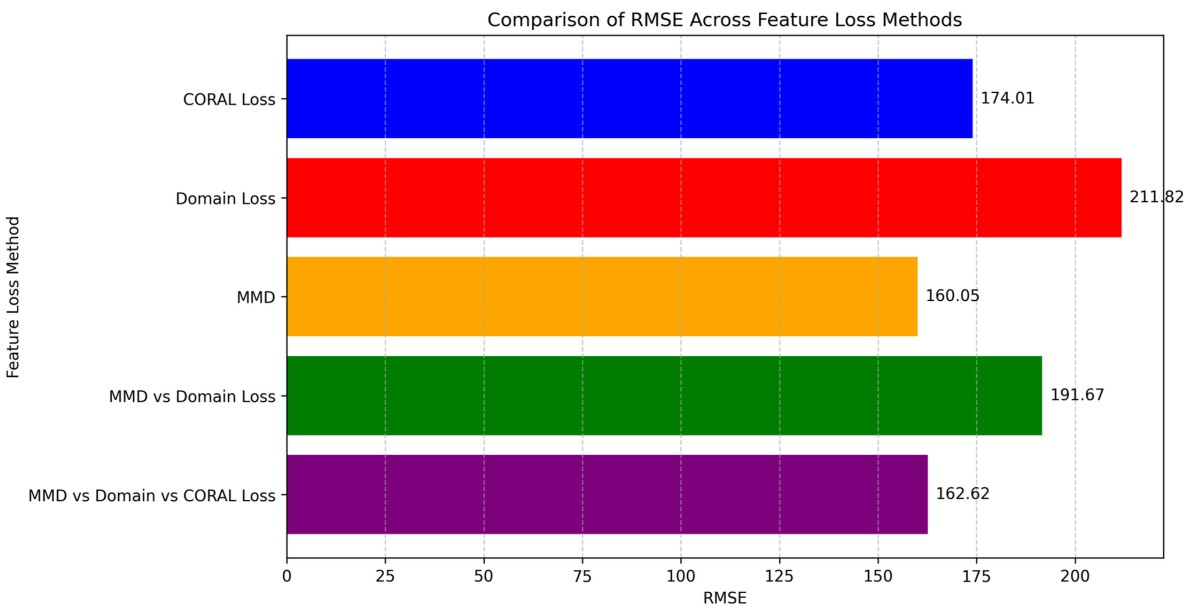

**Fig 7. Comparison of different feature loss methods.** Experiment on the testing data of the second dataset.

in Fig 9, the best performance was achieved when using the second-to-last time step of the Multihead Attention output and a NODE output time step of 2.

The proposed HybridoNet-Adapt model is systematically evaluated under various scenarios by experimenting with four different target sets, each derived from the training data

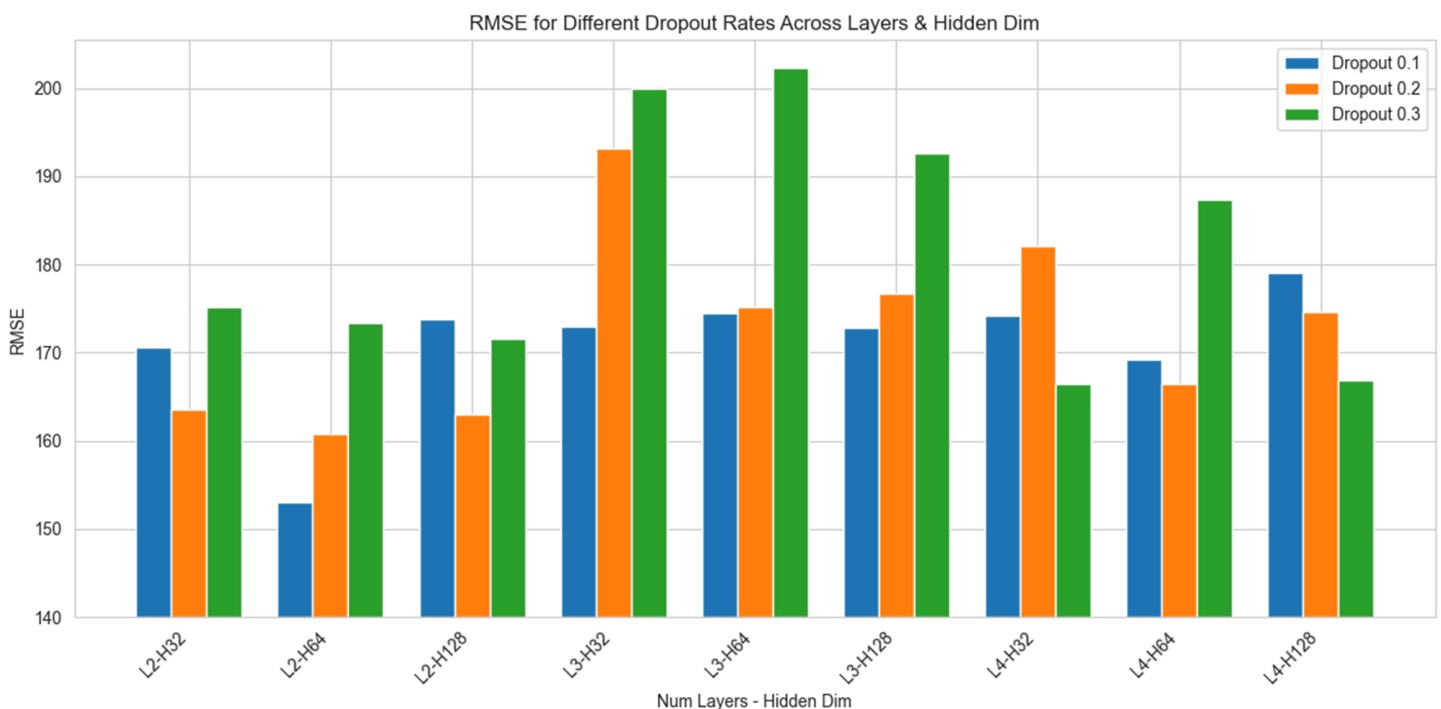

**Fig 8. Comparison of RMSE across different number of LSTM layers, hidden dimensions, and dropout configurations.** Experiment on the testing data of the second dataset.

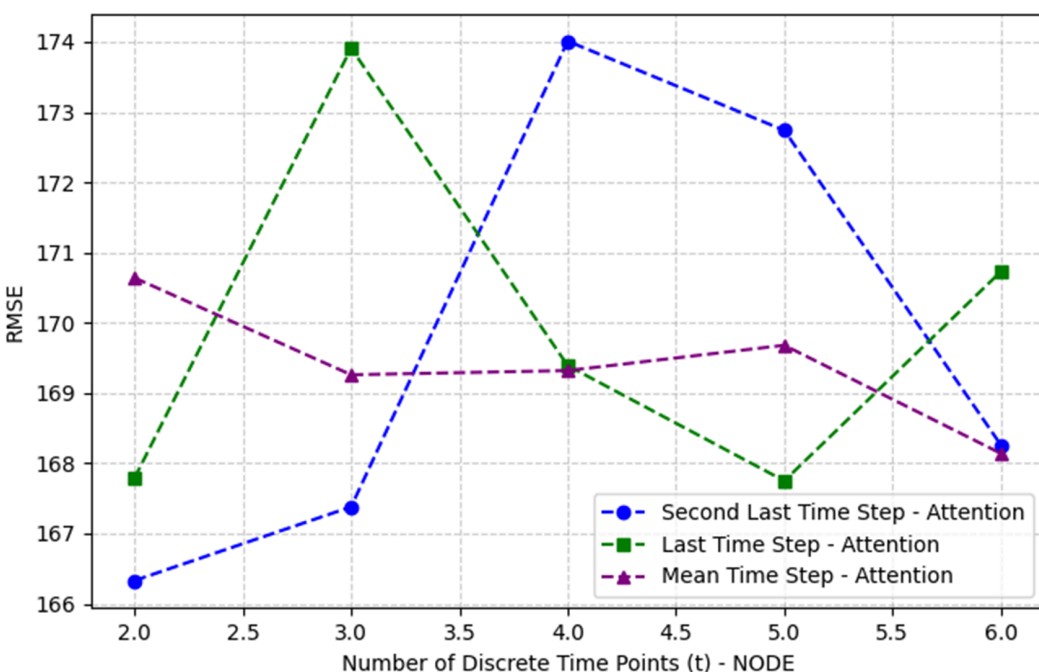

**Fig 9. Comparison of RMSE for different NODE discrete time steps ($t$) and Multihead Attention output time step selections.** Experiment on the testing data of the second dataset.

of the second dataset. The source data is the training data from the first dataset. Below are four target groups of battery cells selected from the training data of the second dataset. These groups are carefully formed to ensure each set represents a diverse range of battery performance. For instance, Group 1 includes both high-cycle cells (e.g., 2-2 with 2,651 cycles) and low-cycle cells (e.g., 1-6 with 1,143 cycles), ensuring a comprehensive representation of aging behaviors.

- **Group 1:** 1-3 (1,858 cycles), 1-6 (1,143 cycles), 2-2 (2,651 cycles), 2-6 (1,572 cycles), 3-2 (2,283 cycles), 3-6 (2,491 cycles), 4-3 (1,142 cycles), 5-4 (1,962 cycles)
- **Group 2:** 1-5 (1,971 cycles), 1-8 (2,285 cycles), 2-4 (1,499 cycles), 2-7 (2,202 cycles), 3-3 (1,649 cycles), 3-7 (2,479 cycles), 4-4 (1,491 cycles), 5-5 (1,583 cycles)
- **Group 3:** 2-8 (1,481 cycles), 3-4 (1,766 cycles), 3-8 (2,342 cycles), 4-1 (2,217 cycles), 4-7 (2,216 cycles), 5-1 (2,507 cycles), 5-6 (2,460 cycles), 6-3 (1,804 cycles)
- **Group 4:** 4-8 (1,706 cycles), 5-2 (1,926 cycles), 5-7 (1,448 cycles), 6-4 (1,717 cycles), 6-5 (2,178 cycles), 7-2 (2,030 cycles), 7-7 (1,685 cycles), 8-2 (2,041 cycles)
- **All:** All battery cells from the training set of the second dataset.

Table 3 shows that HybridoNet-Adapt outperforms both HybridoNet (without DA) and DANN (with DA) across all groups. It achieves the lowest RMSE and MAPE while maintaining the highest $R^2$, demonstrating better generalization. For instance, in Group 1, HybridoNet-Adapt reduces RMSE from 368.99 to 356.46 and improves $R^2$ from 0.21 to 0.30. On the full dataset, it achieves the best RMSE of 153.24 and $R^2$ of 0.88, significantly outperforming DANN, which shows degraded performance (RMSE = 835.35, $R^2$ = –1.37). DANN struggles with large variations in battery aging, while HybridoNet-Adapt effectively adapts to different distributions, leading to consistently better predictions.

Table 4 presents the evaluation metrics for RUL prediction on the test data from the second dataset, comparing Elastic Net, $A_1$, $A_2$ of paper [29] (see Table S4), with our HybridoNet, and HybridoNet-Adapt methods. The results indicate that HybridoNet-Adapt achieves competitive RMSE values, particularly in cases where Elastic Net exhibits high errors. The $R^2$ values show that HybridoNet-Adapt generally improves predictive accuracy compared to the baseline methods. Additionally, MAPE results suggest that HybridoNet-Adapt provides more stable and reliable predictions, especially in challenging scenarios. Overall, these findings demonstrate the potential of HybridoNet-Adapt for enhanced RUL estimation.

Fig 10 illustrates the RUL predictions of XGBoost, HybridoNet, HybridoNet-Adapt, and DANN, compared to the true (observed) RUL for Cell 4-5 and Cell 3-1 in the testing set of the second dataset. Among all methods, HybridoNet-Adapt demonstrates the closest alignment with the observed RUL, highlighting its superior predictive accuracy. This improvement is attributed to HybridoNet-Adapt's ability to align feature representations from

**Table 3**. Comparison of HybridoNet, DANN, and HybridoNet-Adapt across four target data groups from the testing data of the second dataset.

| Group | HybridoNet (Without DA) | | | DANN (With DA) | | | HybridoNet-Adapt (With DA) | | |
|---|---|---|---|---|---|---|---|---|---|
| | RMSE ↓ | R² ↑ | MAPE (%) ↓ | RMSE ↓ | R² ↑ | MAPE (%) ↓ | RMSE ↓ | R² ↑ | MAPE (%) ↓ |
| Group 1 | 368.99 | 0.21 | 18.26 | 604.14 | -0.36 | 27.26 | 356.46 | 0.30 | 17.35 |
| Group 2 | 245.58 | 0.71 | 11.84 | 665.02 | -0.51 | 29.33 | 240.90 | 0.73 | 11.09 |
| Group 3 | 334.79 | 0.35 | 16.69 | 1007.93 | -3.39 | 51.02 | 316.79 | 0.41 | 15.43 |
| Group 4 | 304.91 | 0.61 | 14.08 | 758.16 | -0.99 | 32.84 | 293.40 | 0.63 | 13.27 |
| All | 166.33 | 0.86 | 7.44 | 835,35 | -1,37 | 36,99 | 153.24 | 0.88 | 7.30 |

**Table 4. Evaluation metrics for RUL prediction performance using existing Elastic Net (Ela), $A_1$, $A_2$ results of paper [29] (see Table S4), along with HybridoNet (H), and our proposed HybridoNet-Adapt (H-Adapt).** Experiment on the testing data of the second dataset.

| Channel | RMSE (cycles) | | | | | $R^2$ | | | | | MAPE (%) | | | | |
|---|---|---|---|---|---|---|---|---|---|---|---|---|---|---|---|
| | Ela | $A_1$ | $A_2$ | H | H-Adapt | Ela | $A_1$ | $A_2$ | H | H-Adapt | Ela | $A_1$ | $A_2$ | H | H-Adapt |
| 1-1 | 252 | 63.8 | 42.7 | 30,9 | 57,84 | 0.646 | 0.977 | 0.990 | 0,99 | 0,98 | 14.1 | 3.64 | 2.16 | 1,67 | 3,39 |
| 1-2 | 722 | 262 | 272 | 483,29 | 514,39 | 0.102 | 0.882 | 0.873 | 0,6 | 0,55 | 23.2 | 8.64 | 8.52 | 15,22 | 17,34 |
| 2-5 | 365 | 390 | 364 | 96,9 | 158,42 | 0.122 | / | 0.126 | 0,94 | 0,84 | 19.4 | 23.5 | 22.0 | 5,46 | 9,38 |
| 3-1 | 422 | 104 | 133 | 327,53 | 129,66 | 0.407 | 0.964 | 0.941 | 0,65 | 0,94 | 17.0 | 4.43 | 6.00 | 15,39 | 6,42 |
| 4-5 | 313 | 307 | 301 | 62,55 | 125,98 | 0.493 | 0.512 | 0.531 | 0,98 | 0,92 | 16.1 | 15.7 | 16.3 | 2,69 | 7,4 |
| 5-3 | 757 | 279 | 301 | 346,43 | 392,7 | 0.0225 | 0.867 | 0.845 | 0,8 | 0,74 | 24.2 | 8.94 | 9.47 | 11,18 | 12,4 |
| 6-1 | 310 | 147 | 120 | 70,95 | 34,57 | 0.532 | 0.896 | 0.930 | 0,98 | 0,99 | 14.3 | 7.69 | 6.01 | 3,36 | 1,69 |
| 6-2 | 413 | 96.3 | 141 | 140,41 | 104,43 | 0.415 | 0.968 | 0.932 | 0,93 | 0,96 | 15.5 | 4.08 | 5.98 | 6,37 | 5,03 |
| 6-6 | 672 | 400 | 349 | 226,53 | 248,97 | 0.0821 | 0.675 | 0.753 | 0,9 | 0,87 | 22.4 | 14.5 | 12.6 | 8,71 | 9,52 |
| 6-8 | 609 | 334 | 359 | 148,63 | 217,09 | 0.236 | 0.769 | 0.734 | 0,95 | 0,9 | 20.3 | 12.6 | 13.6 | 5,23 | 8,3 |
| 7-5 | 372 | 58.0 | 46.2 | 68,29 | 142,18 | 0.508 | 0.988 | 0.992 | 0,98 | 0,93 | 15.6 | 2.70 | 2.01 | 2,53 | 6,52 |
| 7-6 | 372 | 295 | 263 | 50,22 | 81,06 | 0.129 | 0.453 | 0.566 | 0,98 | 0,96 | 21.5 | 15.7 | 15.6 | 2,65 | 5 |
| 8-1 | 303 | 200 | 231 | 331,1 | 319,56 | 0.319 | 0.702 | 0.603 | 0,19 | 0,25 | 19.7 | 12.8 | 15.6 | 23,66 | 23,94 |
| 8-5 | 281 | 45.3 | 57.1 | 229,23 | 227,22 | 0.449 | 0.986 | 0.977 | 0,64 | 0,64 | 17.7 | 2.64 | 3.30 | 15,19 | 16,5 |
| 8-6 | 527 | 91.8 | 81.5 | 45,05 | 116,28 | 0.386 | 0.981 | 0.985 | 1 | 0,97 | 18.1 | 3.41 | 2.97 | 1,7 | 4,29 |
| 8-8 | 412 | 363 | 382 | 254,98 | 49,44 | 0.245 | 0.411 | 0.349 | 0,71 | 0,99 | 18.2 | 18.2 | 19.3 | 13,3 | 2,31 |
| 9-4 | 431 | 104 | 74.3 | 102,13 | 34,23 | 0.406 | 0.966 | 0.982 | 0,97 | 1 | 16.9 | 4.56 | 3.17 | 3,79 | 1,43 |
| 9-6 | 403 | 297 | 292 | 182,32 | 56,22 | 0.328 | 0.634 | 0.649 | 0,86 | 0,99 | 18.4 | 15.4 | 15.1 | 6,24 | 2,97 |
| 10-1 | 386 | 111 | 81.0 | 27,28 | 57,33 | 0.355 | 0.947 | 0.972 | 1 | 0,99 | 17.6 | 5.03 | 4.24 | 1,28 | 2,52 |
| 10-4 | 331 | 67.5 | 60.9 | 184,35 | 43,13 | 0.582 | 0.983 | 0.986 | 0,87 | 0,99 | 14.3 | 3.15 | 2.95 | 8,36 | 2,05 |
| 10-6 | 485 | 112 | 84.0 | 188,2 | 103,52 | 0.442 | 0.970 | 0.983 | 0,92 | 0,97 | 16.0 | 3.93 | 3.12 | 6,62 | 3,96 |
| 10-7 | 405 | 86.7 | 52.8 | 62,06 | 157,16 | 0.353 | 0.970 | 0.989 | 0,98 | 0,9 | 16.7 | 3.25 | 1.97 | 3,13 | 8,28 |
| **Mean** | 434 | 192 | 186 | 166,33 | 153,24 | 0.344 | 0.795 | 0.804 | 0,86 | 0,88 | 18.1 | 8.84 | 8.72 | 7,44 | 7,3 |

the source domain to the target domain, as shown in Fig 11. By effectively increasing the amount of target-relevant data through our domain adaptation technique, HybridoNet-Adapt enhances robustness, making it more adaptable to diverse real-world battery degradation scenarios.

## 4.8 Comparison with state-of-the-art methods

Fig 12a presents a performance comparison of different models on the secondary testing data from the first dataset. The Multi-Time Scale Feature Extraction Hybrid (MSFEH) model [42], XGBoost, and HybridoNet were trained using the training data from the first dataset. HybridoNet-Adapt, in contrast, was trained with the training data of the second dataset as the source input and the training data of the first dataset as the target input. HybridoNet-Adapt achieves the best results, with the lowest RMSE (146.52), demonstrating its superior predictive accuracy through domain adaptation. Moreover, HybridoNet outperforms both XGBoost and MSFEH, highlighting the effectiveness of deep learning–based approaches. The additional improvements achieved by HybridoNet-Adapt further validate the benefits of domain adaptation in enhancing RUL prediction performance. It should be noted that in the MSFEH paper [42], the MAPE formula differs from the one used in our work; therefore, we report only the RMSE comparison with MSFEH.

Fig 12b presents a comparison of our HybridoNet and HybridoNet-Adapt models with state-of-the-art methods, including Elastic Net [29], $A_1$ [29], $A_2$ [29], Ridge Linear [23], Random Forest [23], and Structural Pruning [66], evaluated on the testing data from the second dataset. HybridoNet-Adapt, was trained with the training data of the first dataset as the source input and the training data of the second dataset as the target input, whereas the other

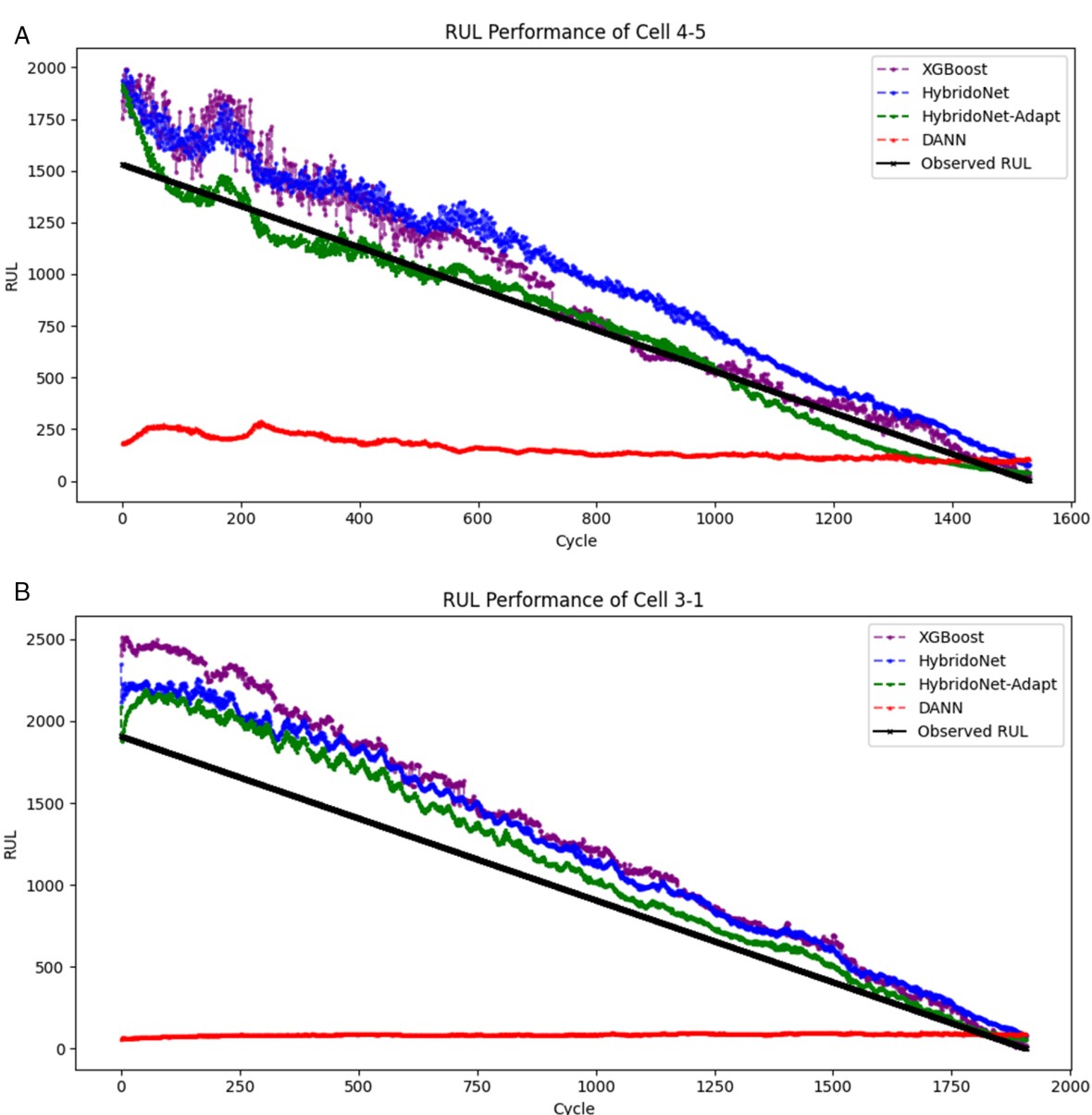

**Fig 10. Comparison of model predictions with observed RUL for Cells 4-5 and 3-1 from the testing data of the second dataset.**

methods were trained solely on the training data from the second dataset. The results demonstrate that HybridoNet-Adapt achieves the lowest RMSE (153.24), outperforming all other approaches. This highlights the effectiveness of our proposed method in enhancing predictive performance. Overall, HybridoNet-Adapt consistently outperforms across large datasets with diverse charging and discharging profiles.

In future work, we plan to explore Physics-Informed Neural Networks (PINNs) as a approach to enhance interpretability and physical consistency. By embedding governing equations and domain-specific constraints directly into the learning process, PINNs can reduce

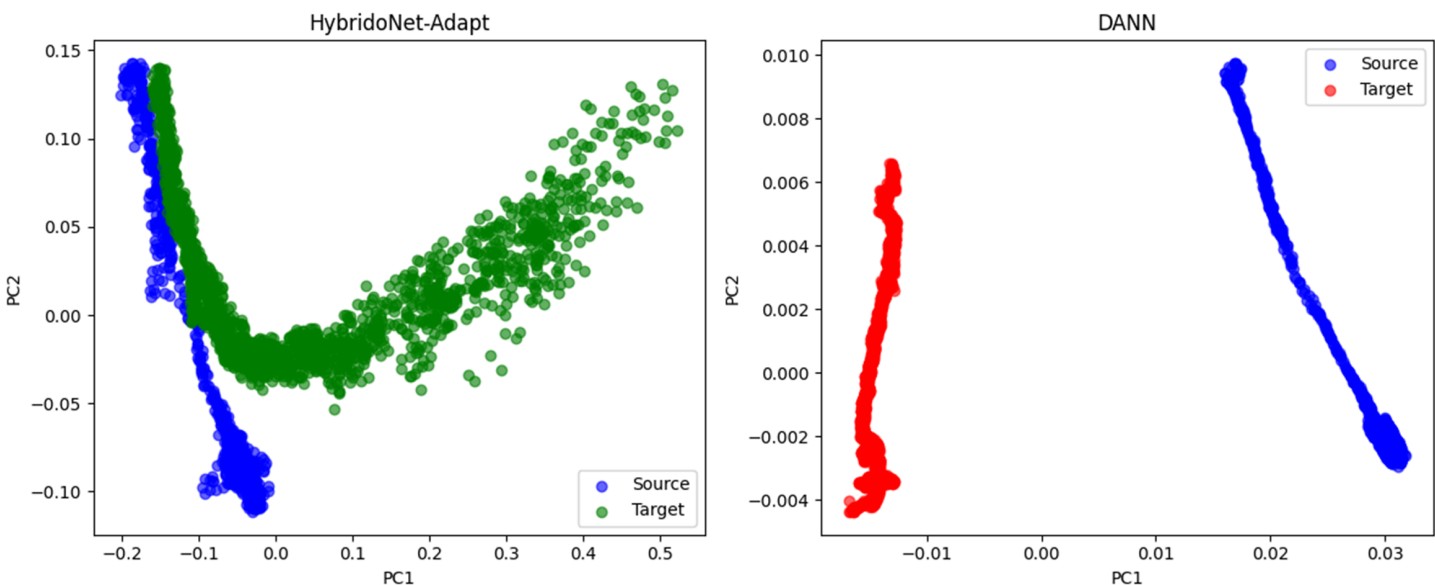

**Fig 11. PCA-based comparison of embedding features between HybridoNet-Adapt and DANN from Cell 3-1 the testing data of the second dataset.**

reliance on purely data-driven correlations while improving extrapolation to unseen conditions. Such a framework would allow the model to not only achieve strong predictive performance but also provide insights grounded in physical principles, thereby addressing the gap between predictive accuracy and practical interpretability.

## 5 Conclusion

In this paper, we proposed *HybridoNet-Adapt*, a novel domain-adaptive framework for accurate Remaining Useful Life (RUL) prediction of lithium-ion batteries. Our approach addresses the challenge of distribution shift between training and testing data by leveraging domain adaptation techniques, specifically Maximum Mean Discrepancy (MMD), to align feature representations between source and target domains. By integrating a hybrid prediction mechanism with trainable trade-off parameters, the model effectively balances contributions from both domain-specific predictors. The proposed architecture combines LSTM, Multi-head Attention, and NODE blocks within a feature extractor, enabling the model to capture both temporal and dynamic characteristics of battery degradation. Extensive experiments on two large-scale benchmark datasets demonstrate that *HybridoNet-Adapt* consistently outperforms state-of-the-art baselines, such as XGBoost and Elastic Net, as well as state-of-the-art deep learning models like Structural Pruning and Multi-Time Scale Feature Extraction Hybrid (MSFEH) model, affirming its effectiveness in the RUL prediction task. For future work, we plan to enhance model generalization through Physics-Informed Neural Networks, and explore multi-modal data integration to improve scalability and robustness across diverse BHM applications. We will investigate the integration of multi-modal data to enhance scalability and robustness across a wide range of BHM applications.

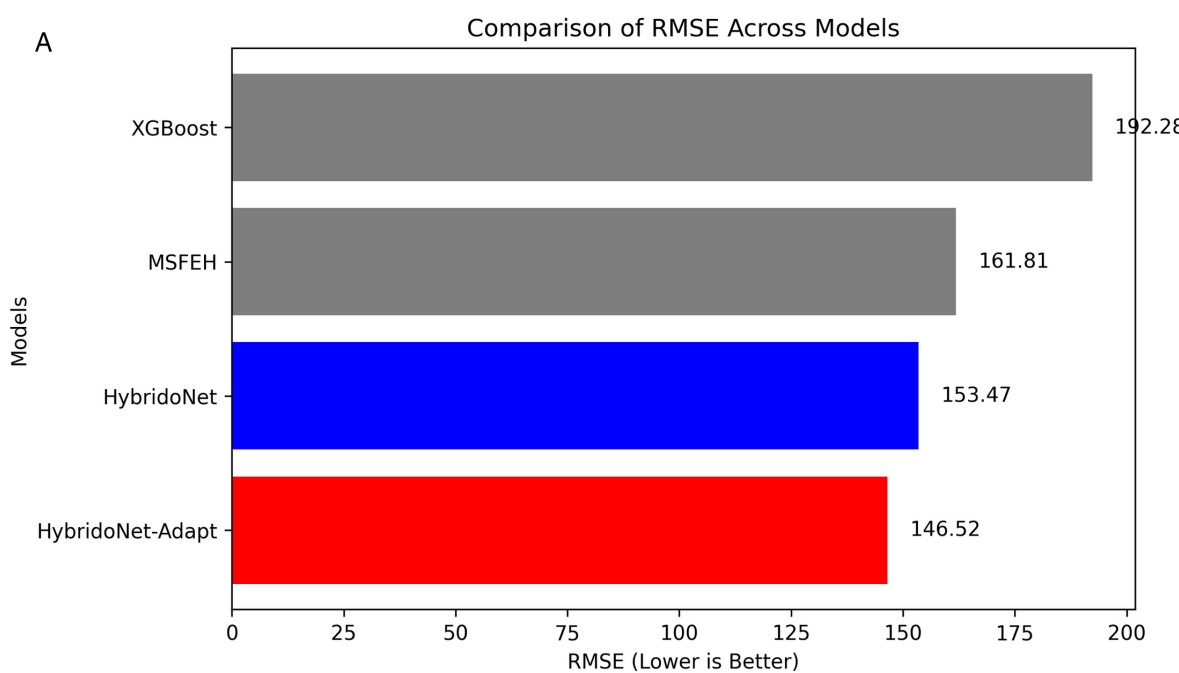

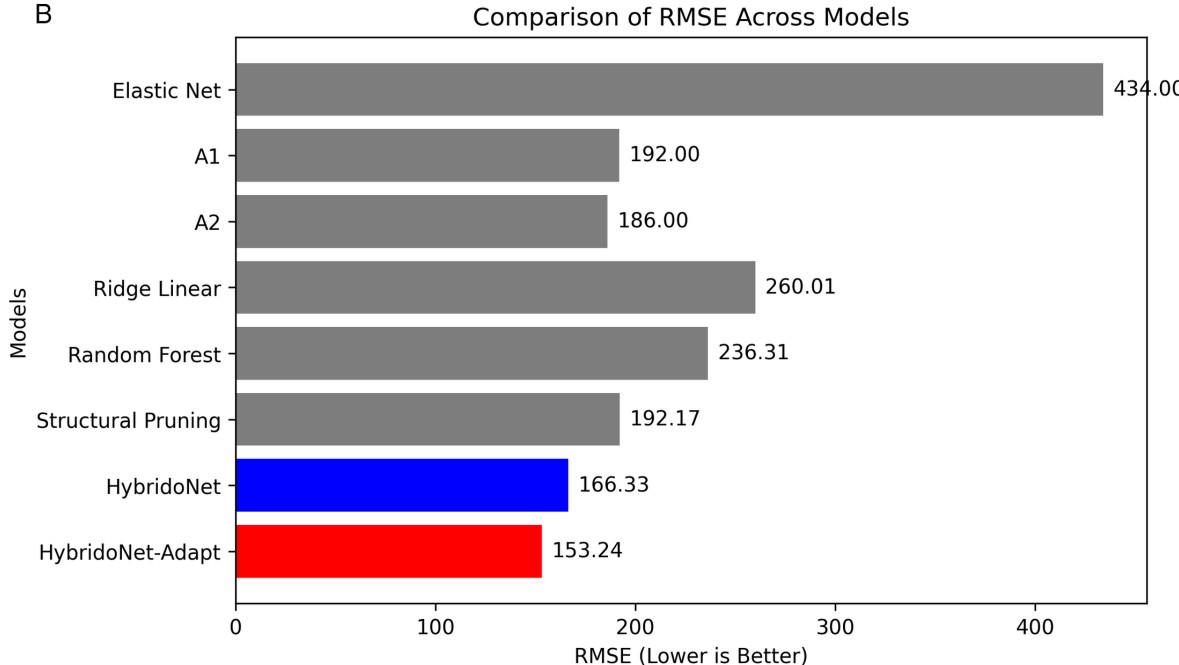

**Fig 12. Comparison of our proposed models with existing state-of-the-art methods.** (a) Results on the secondary testing data of the first dataset. (b) Results on the testing data of the second dataset. (a) The first dataset. (b) The second dataset.

## Preprint availability

A preprint of this manuscript is available at: https://arxiv.org/pdf/2503.21392.

## Contact information

For access to the code and further information about this proposed system, please contact AIWARE Limited Company at: https://aiware.website/Contact.

## Author contributions

**Conceptualization:** Hung-Cuong Trinh.

**Data curation:** Lam Pham, Duong Tran Anh.

**Formal analysis:** Khoa Tran.

**Funding acquisition:** Hung-Cuong Trinh.

**Investigation:** Khoa Tran, Bao Huynh, Tri Le, Vy-Rin Nguyen.

**Methodology:** Khoa Tran, Bao Huynh, Tri Le, Vy-Rin Nguyen.

**Project administration:** Hung-Cuong Trinh.

**Resources:** Khoa Tran.

**Software:** Vy-Rin Nguyen.

**Supervision:** Hung-Cuong Trinh.

**Validation:** Tri Le, Lam Pham, Duong Tran Anh.

**Visualization:** Bao Huynh, Lam Pham, Duong Tran Anh.

**Writing – original draft:** Khoa Tran, Bao Huynh, Tri Le, Vy-Rin Nguyen, Hung-Cuong Trinh.

**Writing – review & editing:** Khoa Tran, Bao Huynh, Tri Le, Vy-Rin Nguyen, Hung-Cuong Trinh.

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
