## [Decision Letter · Decision Letter 0]

10 Sep 2025

PONE-D-25-42933HybridoNet-Adapt: A Domain-Adapted Framework for Accurate Lithium-Ion Battery RUL PredictionPLOS ONE

Dear Dr. Trinh,

Thank you for submitting your manuscript to PLOS ONE. After careful consideration, we feel that it has merit but does not fully meet PLOS ONE’s publication criteria as it currently stands. Therefore, we invite you to submit a revised version of the manuscript that addresses the points raised during the review process.

We look forward to receiving your revised manuscript.

Kind regards,

Zhibin Zhao

Academic Editor

PLOS ONE

Journal Requirements:

5. Please include a new copy of Table 1 and 3 in your manuscript; the current table is difficult to read. Please follow the link for more information: https://blogs.plos.org/plos/2019/06/looking-good-tips-for-creating-your-plos-figures-graphics/"

Additional Editor Comments :

Reviewer #1:

Overall, the manuscript presents an interesting framework (HybridoNet-Adapt) that combines domain adaptation with deep temporal feature extraction for lithium-ion battery RUL prediction. The idea of addressing domain shift through Maximum Mean Discrepancy (MMD)-based alignment and hybrid predictors is promising. However, several important issues remain that limit the clarity and rigor of the work. The following points should be addressed to further improve the manuscript:

1. On Page 3, the statement “Many existing methods predict RUL based on estimated maximum discharge capacity, representing the remaining life as a percentage of the nominal capacity” is actually the definition of SOH, not RUL. Please carefully distinguish between these two concepts. Accordingly, some references listed in Table 1 are not truly RUL prediction studies.

2. To further strengthen the introduction, it would be beneficial to include more recent and relevant works, such as DOI: 10.1016/j.jpowsour.2022.230975; 10.1016/j.est.2025.116024.

3. The manuscript currently lists five contributions. This number is excessive and somewhat redundant. It would be clearer and more impactful to condense and reorganize them into three key contributions.

4. Section titles should not contain references or colons. Please revise accordingly.

5. All equations should be numbered, and every variable used in the equations must be explained. Avoid using whole words as variable names in formulas.

6. In Figure 2, the arrows pointing from the MSE loss and MMD loss to the Input are confusing. Please check whether the statement is correct or explain the meaning of the arrows.

7. It appears that target-domain labels may have been used from Page 7. If so, why not directly train and test using the target-domain data? Please clarify whether target labels are available, and if they are, discuss the meaningfulness of the domain adaptation task under this setting.

8. In many published works using the TRI dataset, models are typically trained on Batch 1 and tested on Batches 2 and 3. If target-domain labels were used to aid model training, then the task defined in this paper may be easier, and the results may not provide strong evidence that the proposed method is more effective than existing approaches.

9. Hyperparameters should be presented in a table for clarity and reproducibility.

Reviewer #2:

This paper proposes HybridoNet-Adapt, a novel domain-adaptive framework for accurate Remaining Useful Life (RUL) prediction of lithium-ion batteries. Although the experimental results seem promising, there are still some major issues. Therefore, the paper needs to be rejected and resubmitted.

Other comments:

1) The study lacks practical interpretability. Although the methodology is emphasized, the model predictions are not supported by physical explanations or visual analyses.

2) The baselines are mostly traditional methods or relatively simple deep learning models, while more recent approaches such as advanced transfer learning or self-supervised learning are not systematically compared.

3) While different feature extraction modules and alignment losses are examined, the ablation study remains incomplete, as it does not explore alternative feature selection strategies or the sensitivity to different sampling window lengths.

Reviewers' comments:

Reviewer's Responses to Questions

**Comments to the Author**

1. Is the manuscript technically sound, and do the data support the conclusions?

Reviewer #1: Partly

Reviewer #2: Yes

2. Has the statistical analysis been performed appropriately and rigorously? 

Reviewer #1: No

Reviewer #2: Yes

3. Have the authors made all data underlying the findings in their manuscript fully available?

Reviewer #1: Yes

Reviewer #2: Yes

4. Is the manuscript presented in an intelligible fashion and written in standard English?

Reviewer #1: Yes

Reviewer #2: Yes

5. Review Comments to the Author

Reviewer #1: Overall, the manuscript presents an interesting framework (HybridoNet-Adapt) that combines domain adaptation with deep temporal feature extraction for lithium-ion battery RUL prediction. The idea of addressing domain shift through Maximum Mean Discrepancy (MMD)-based alignment and hybrid predictors is promising. However, several important issues remain that limit the clarity and rigor of the work. The following points should be addressed to further improve the manuscript:

1. On Page 3, the statement “Many existing methods predict RUL based on estimated maximum discharge capacity, representing the remaining life as a percentage of the nominal capacity” is actually the definition of SOH, not RUL. Please carefully distinguish between these two concepts. Accordingly, some references listed in Table 1 are not truly RUL prediction studies.

2. To further strengthen the introduction, it would be beneficial to include more recent and relevant works, such as DOI: 10.1016/j.jpowsour.2022.230975; 10.1016/j.est.2025.116024.

3. The manuscript currently lists five contributions. This number is excessive and somewhat redundant. It would be clearer and more impactful to condense and reorganize them into three key contributions.

4. Section titles should not contain references or colons. Please revise accordingly.

5. All equations should be numbered, and every variable used in the equations must be explained. Avoid using whole words as variable names in formulas.

6. In Figure 2, the arrows pointing from the MSE loss and MMD loss to the Input are confusing. Please check whether the statement is correct or explain the meaning of the arrows.

7. It appears that target-domain labels may have been used from Page 7. If so, why not directly train and test using the target-domain data? Please clarify whether target labels are available, and if they are, discuss the meaningfulness of the domain adaptation task under this setting.

8. In many published works using the TRI dataset, models are typically trained on Batch 1 and tested on Batches 2 and 3. If target-domain labels were used to aid model training, then the task defined in this paper may be easier, and the results may not provide strong evidence that the proposed method is more effective than existing approaches.

9. Hyperparameters should be presented in a table for clarity and reproducibility.

Reviewer #2: This paper proposes HybridoNet-Adapt, a novel domain-adaptive framework for accurate Remaining Useful Life (RUL) prediction of lithium-ion batteries. Although the experimental results seem promising, there are still some major issues. Therefore, the paper needs to be rejected and resubmitted.

Other comments:

1) The study lacks practical interpretability. Although the methodology is emphasized, the model predictions are not supported by physical explanations or visual analyses.

2) The baselines are mostly traditional methods or relatively simple deep learning models, while more recent approaches such as advanced transfer learning or self-supervised learning are not systematically compared.

3) While different feature extraction modules and alignment losses are examined, the ablation study remains incomplete, as it does not explore alternative feature selection strategies or the sensitivity to different sampling window lengths.

6. PLOS authors have the option to publish the peer review history of their article (what does this mean?). If published, this will include your full peer review and any attached files.

Reviewer #1: No

Reviewer #2: No

---

## [Author Response · Author response to Decision Letter 1]

22 Sep 2025

Response Letter

We sincerely thank Editor-in-Chief, Associate Editor, and the reviewers for their constructive,

pertinent, and useful comments on our manuscript.

To Editor-in-Chief and Associate Editor

We are pleased to see insightful and constructive comments from the reviewers. In this resubmission, we have attempted to address the reviewers’ concerns and revised the manuscript

accordingly. Further details about all the changes we made can be found in our point-to-point

response to the reviewers’ comments below. In the revised manuscript, we marked the changes

in yellow.

To Reviewer 1

On Page 3, the statement "Many existing methods predict RUL based on estimated maximum

discharge capacity, representing the remaining life as a percentage of the nominal capacity" is

actually the definition of SOH, not RUL. Please carefully distinguish between these two concepts.

Accordingly, some references listed in Table 1 are not truly RUL prediction studies.

Response: Thank you for your comment. We have revised the statement to reflect the RUL

concept. The updated text begins with "Historical data-independent methods estimate the

future capacity trajectory ..." on Page 3 and "RUL can be assessed based on the number of

remaining cycles" on Page 2. In addition, we have removed a reference that predicts cycle life

rather than RUL from Table 1.

Comment 2: To further strengthen the introduction, it would be beneficial to include more recent

and relevant works, such as DOI: 10.1016/j.jpowsour.2022.230975; 10.1016/j.est.2025.116024.

Response: Thank you for your comment. We have added the two references by updating Table

1 and including a paragraph beginning with "[31] proposed a transferable RUL prediction ..." in

the Problem Statement and Potential section.

Comment 3: The manuscript currently lists five contributions. This number is excessive and

somewhat redundant. It would be clearer and more impactful to condense and reorganize them

into three key contributions.

Response: Thank you for your comment. We have updated the contributions and condensed them

into three key points. The revised text begins with “We propose a historical data-independent

RUL prediction framework for . . . ”.

Comment 4: Section titles should not contain references or colons. Please revise accordingly.

1

Response: Thank you for your comment. We have remove the references and colons from two

section titles (LSTM, Multihead Attention, NODE).

Comment 5: All equations should be numbered, and every variable used in the equations must

be explained. Avoid using whole words as variable names in formulas.

Response: Thank for your comment. We have revised the equations in the Preliminaries section,

ensuring that all equations are numbered and every variable is clearly defined, while avoiding

the use of whole words as variable names.

Comment 6: In Figure 2, the arrows pointing from the MSE loss and MMD loss to the Input are

confusing. Please check whether the statement is correct or explain the meaning of the arrows.

Response: Thank for your comment. We have revised Figure 2 by removing the arrows from

the MSE and MMD loss blocks to the input, ensuring that the diagram accurately reflects the

intended training flow.

Comment 7: It appears that target-domain labels may have been used from Page 7. If so, why

not directly train and test using the target-domain data? Please clarify whether target labels

are available, and if they are, discuss the meaningfulness of the domain adaptation task under

this setting.

Response: Thank you for your comment. Yes, target-domain labels are available. To address this,

we created HybridoNet (without domain adaptation), which consists of only a feature extractor

and a target predictor and is trained solely on labeled target data. The updated text, beginning

with "For validating the proposed domain adaptation used in HybridoNet-Adapt, we ...", has

been added to the Proposed method / HybridoNet-Adapt: A Proposed RUL Prediction model

with Novel Domain Adaptation section.

Comment 8: In many published works using the TRI dataset, models are typically trained on

Batch 1 and tested on Batches 2 and 3. If target-domain labels were used to aid model training,

then the task defined in this paper may be easier, and the results may not provide strong evidence

that the proposed method is more effective than existing approaches.

Response: Thank you for your comment. We have added Figure 11(a), which compares our proposed HybridoNet-Adapt (146.52 RMSE) with the MSFEH 2025 state-of-the-art model (161.81

RMSE). This result shows that our proposed method, even when using target-domain labels,

substantially outperforms the current state-of-the-art under this setting.

For the second dataset, we have removed the comparison with the Dual-input DNN because it

2

was conducted under a different experimental setup. Our proposed method follows the official

split of the second dataset, with 55 LIBs for training and 22 LIBs for testing. In contrast, the

Dual-input DNN work re-split the dataset, using 70 batteries for training and only 7 for testing,

which makes its evaluation less generalizable.

Comment 9: Hyperparameters should be presented in a table for clarity and reproducibility.

Response: Thank you for your comment. We have included Table 2 for the hyperparameters,

along with a paragraph explaining the notation, beginning with "The hyperparameters of the

proposed HybridoNet-Adapt are summarized ...".

To Reviewer 2

Comment 1: The study lacks practical interpretability. Although the methodology is emphasized,

the model predictions are not supported by physical explanations or visual analyses.

Response: Thank you for your comment. We acknowledge the importance of interpretability in

bridging the gap between predictive performance and physical insights. We plan to integrate

Physics-Informed Neural Networks in future work, as noted in the added text beginning with

“In future work, we plan to explore Physics-Informed Neural Networks (PINNs) . . . ” in the

Experiments and discussion Section.

Comment 2: The baselines are mostly traditional methods or relatively simple deep learning models, while more recent approaches such as advanced transfer learning or self-supervised learning

are not systematically compared.

Response: Thank you for your comment. We have compared our model with recent state-of-theart approaches. For the first dataset, we have added Figure 11(a), which compares our proposed

HybridoNet-Adapt (146.52 RMSE) with the MSFEH 2025 state-of-the-art model (161.81 RMSE).

This result shows that our proposed method substantially outperforms the current state-of-theart under this setting.

For the second dataset, we have added the Structural Pruning 2024 state-of-the-art model in

Figure 11(b). We also removed the comparison with the Dual-input DNN because it was conducted under a different experimental setup. Our proposed method follows the official split of

the second dataset, with 55 LIBs for training and 22 LIBs for testing. In contrast, the Dual-input

DNN re-split the dataset, using 70 batteries for training and only 7 for testing, which makes its

evaluation less generalizable.

Comment 3: While different feature extraction modules and alignment losses are examined, the

ablation study remains incomplete, as it does not explore alternative feature selection strategies

or the sensitivity to different sampling window lengths.

Response: Thank you for your comment. We have added an ablation study (see Figure 5 and the

accompanying text beginning with "Figure 5a presents the feature importance ranking derived

3

from XGBoost...") to compare our chosen statistical feature extraction method with alternative

approaches.

4

---

## [Decision Letter · Decision Letter 1]

6 Oct 2025

HybridoNet-Adapt: A Domain-Adapted Framework for Accurate Lithium-Ion Battery RUL Prediction

PONE-D-25-42933R1

Dear Dr. Trinh,

We’re pleased to inform you that your manuscript has been judged scientifically suitable for publication and will be formally accepted for publication once it meets all outstanding technical requirements.

Kind regards,

Zhibin Zhao

Academic Editor

PLOS ONE

Additional Editor Comments (optional):

The authors have well sovled all the comments from the reviewers.

Reviewers' comments:

Reviewer's Responses to Questions

**Comments to the Author**

1. If the authors have adequately addressed your comments raised in a previous round of review and you feel that this manuscript is now acceptable for publication, you may indicate that here to bypass the “Comments to the Author” section, enter your conflict of interest statement in the “Confidential to Editor” section, and submit your "Accept" recommendation.

Reviewer #1: All comments have been addressed

Reviewer #2: All comments have been addressed

2. Is the manuscript technically sound, and do the data support the conclusions?

Reviewer #1: Partly

Reviewer #2: Yes

3. Has the statistical analysis been performed appropriately and rigorously? 

Reviewer #1: I Don't Know

Reviewer #2: Yes

4. Have the authors made all data underlying the findings in their manuscript fully available?

Reviewer #1: Yes

Reviewer #2: Yes

5. Is the manuscript presented in an intelligible fashion and written in standard English?

Reviewer #1: Yes

Reviewer #2: Yes

6. Review Comments to the Author

Reviewer #1: The authors have provided satisfactory and detailed responses to all of my previous comments, and I have no further questions or concerns at this stage.

Reviewer #2: I have no further questions, and the manuscript can be considered for acceptance....................

7. PLOS authors have the option to publish the peer review history of their article (what does this mean?). If published, this will include your full peer review and any attached files.

Reviewer #1: No

Reviewer #2: No

---

## [Editor Report · Acceptance letter]

PONE-D-25-42933R1

PLOS ONE

Dear Dr. Trinh,

I'm pleased to inform you that your manuscript has been deemed suitable for publication in PLOS ONE. Congratulations! Your manuscript is now being handed over to our production team.

Kind regards,

on behalf of

Dr. Zhibin Zhao

Academic Editor

PLOS ONE